# 3D Gaussian Splatting as Markov Chain Monte Carlo

**Shakiba Kheradmand[1], Daniel Rebain[1], Gopal Sharma[1],**
**Weiwei Sun[1], Yang-Che Tseng[1], Hossam Isack[2], Abhishek Kar[2],**
**Andrea Tagliasacchi[3, 4, 5], Kwang Moo Yi[1]**

[1]University of British Columbia, [2]Google Research,
[3]Google DeepMind, [4]Simon Fraser University, [5]University of Toronto

https://ubc-vision.github.io/3dgs-mcmc

## Abstract

While 3D Gaussian Splatting has recently become popular for neural rendering, current methods rely on carefully engineered cloning and splitting strategies for placing Gaussians, which can lead to poor-quality renderings, and reliance on a good initialization. In this work, we rethink the set of 3D Gaussians as a random sample drawn from an underlying probability distribution describing the physical representation of the scene—in other words, Markov Chain Monte Carlo (MCMC) samples. Under this view, we show that the 3D Gaussian updates can be converted as Stochastic Gradient Langevin Dynamics (SGLD) update by simply introducing noise. We then rewrite the densification and pruning strategies in 3D Gaussian Splatting as simply a deterministic state transition of MCMC samples, removing these heuristics from the framework. To do so, we revise the 'cloning' of Gaussians into a relocalization scheme that approximately preserves sample probability. To encourage efficient use of Gaussians, we introduce a regularizer that promotes the removal of unused Gaussians. On various standard evaluation scenes, we show that our method provides improved rendering quality, easy control over the number of Gaussians, and robustness to initialization.

## 1 Introduction

Neural Radiance Fields (NeRF) have been at the forefront of these advancements, providing impressive results through implicit modeling via neural networks, the landscape is further evolving with the introduction of 3D Gaussian splatting.

Neural rendering has seen a significant advancement with the introduction of Neural Radiance Fields (NeRF) [28], and more recently, 3D Gaussian Splatting (3DGS) [19]. 3D Gaussian Splatting became highly popular thanks to its speed and efficiency—it can render high-quality images in a fraction of the time required by NeRF. Unsurprisingly, various extensions to 3D Gaussian Splatting have been proposed, such as extending 3D Gaussians to dynamic scenes [43, 41], making them robust to aliasing effects [48, 42], and generative 3D content creation [52, 49, 36].

However, despite the various extensions, a common shortcoming of these methods is that they mostly rely on the same initialization and densification strategy for placing the Gaussians: either the one originally suggested by [19], or other recently proposed variations [4]. Specifically, they rely on carefully engineered cloning and splitting heuristics for placing Gaussians [19, see "adaptive density control"]. Depending on the state of each Gaussian, they are cloned, split, or pruned, which is the primary way to control the number of Gaussians within the 3DGS representation. Moreover,

38th Conference on Neural Information Processing Systems (NeurIPS 2024).

Gaussians are regularly 'reset' by setting their opacities to small values to remove *floaters* in the representations. This heuristic-based approach requires multiple hyperparameters to be carefully tuned and, as we will show later on in the paper, can fail in some scenes.

These heuristics further lead to various problems. It causes the method to heavily rely on good initial point clouds for it to work well, especially when applied to real-world scenes. It is also nontrivial to estimate how many 3D Gaussians will be used for a given scene from just the hyperparameters, making it difficult to control the computation and memory budget in advance without affecting reconstruction quality during inference time. Concurrent works [10, 9] thus focus on having better initialization to solve the former, or study the latter problem [4]. However, even these recent concurrent solutions still rely on heuristics, and they do not always generalize well. In some cases, this leads to sub-optimal placement of Gaussians resulting in poor quality renderings and wasted compute.

To solve this problem, we take a step back and rethink the set of 3D Gaussians as random samples—more specifically, as Markov Chain Monte Carlo (MCMC) samples—drawn from an underlying probability distribution that is proportional to how faithfully these Gaussians reconstruct the scene. With this considered, we show in Section 3.2 that the conventional 3D Gaussian Splatting update rule is very similar to a Stochastic Gradient Langevin Dynamics update [3, 20], where the only term missing is a noise term that promotes the exploration of samples. We thus reformulate 3D Gaussian Splatting into an SGLD framework, an MCMC algorithm, which naturally explores the scene landscape, and samples Gaussians that are effective in reproducing the scene faithfully. It is important to note that while we assume an underlying distribution, this distribution does not need to be explicitly modelled, as the Gaussians themselves are already representing it.

Given this view, the heuristics involved in the densification and pruning of Gaussians, as well as resetting their opacities, are no longer necessary. 3D Gaussians are simply the samples used for MCMC, thus exploring their sample locations is naturally dealt through SGLD updates. Any modification to the set of Gaussians, including increasing or decreasing their cardinality, can be reformulated as a deterministic state transition—relocation of Gaussians—where we move our sample (the set of Gaussians) to another sample (another set of Gaussians with different configuration). Importantly, to minimally disturb the MCMC sampling chain, we make sure that the two states, before and after the move, are of similar probability. This implies that the training loss value does not change as we alter combinatorial properties, such as the number of Gaussians within the representation, preventing the training process from becoming unstable. This simple strategy, under the MCMC framework, is enough to provide renderings of high quality, beyond what is provided by the conventional heuristics.

In more detail, we propose to relocate Gaussians using a 'cloning' strategy, where we move 'dead' Gaussians (with low opacity) to where other 'live' Gaussians exist, but in a way that has minimal impact on the rendering, and thus on the probability distribution of the Gaussians. In other words, we set the composition of the cloned Gaussians to render the same images as before cloning. While a modification of cloning [4] was recently suggested as a way to ensure the rendering is equal at the Gaussian centers, we show that this is not enough—the whole Gaussian must be considered. Without our careful strategy MCMC sampling provides sub-optimal training. Finally, to encourage efficient use of the Gaussians, we apply L1-regularization. As the extent of a Gaussian is defined both the opacity and the scale of Gaussians, we apply our regularization to both of them. This effectively encourages them to 'disappear' if unnecessary.

We evaluate our method on standard scenes evaluated in [19] (NeRF Synthetic [28], MipNeRF 360 [2], Tank & Temples [22], Deep Blending [16]), as well as the OMMO [27] dataset that exhibit large scene context. With our method, one does not need to initialize the Gaussians carefully. Our method provides high-quality renderings, regardless of whether Gaussians are initialized randomly or from Structure-from-Motion points.

To summarize, our contributions are:

- we reveal the link between 3DGS and MCMC sampling, leading to a simpler optimization;
- we replace the heuristics in 3D Gaussian Splatting with a principled relocation strategy;
- we introduce regularizer to encourage parsimonious use of Gaussians;
- we improve robustness to initialization;
- we provide higher rendering quality.

## 2  Related Work

**Novel-view synthesis via Neural Radiance Fields.** Since the introduction of Neural Radiance Fields (NeRF) [28], it has become extremely popular for building and representing 3D scenes. The core idea behind the method is to learn a neural field that encodes radiance values in the modeling volume, which is then used to render via volume rendering with light rays. Since its first introduction, it has been extended to deal with few views [47], to generalize to new scenes without training [34, 47], to dynamic [12] and unbounded scenes [2], to roughly posed images [39], to speed up training [29, 46], and even to biomedical applications [7, 51] to name a few. These extensions are by no means an exhaustive list and demonstrate the impact that NeRF had. For a more in-depth survey we refer the readers to [13].

Amongst works on NeRF, most relevant to ours is [20], which also employs Stochastic Gradient Langevin Dynamics (SGLD) [3] to identify the most promising samples to train with, and allow faster training convergence. While we ground ourselves in the same Markov Chain Monte Carlo (MCMC) paradigm based on SGLD, the application context is entirely different. In their work, SGLD is used to perform a form of 'soft mining', so to accelerate NeRF training. In our case, we are instead rethinking 3DGS as samples from an underlying distribution that represents the 3D scene.

**Gaussian Splatting.** 3D Gaussian Splatting (3DGS) [19] is a recent alternative to NeRF that rely on differentiable rasterization instead of volume rendering. In a nutshell, instead of querying points along the ray, it stores Gaussians, which can then be rasterized into each view to form images. This rasterization operation is highly efficient, as instead of querying hundreds of points along a light ray to render a pixel, one can simply rasterize the (few) Gaussians associated with a given pixel. Therefore, Gaussian Splatting allows 1080p images to be rendered at 130 frames per second on modern GPUs, which catalyzed the research community.

Unsurprisingly, various extensions immediately followed. These include ones that focus on removing aliasing effects [48, 42], allowing reflection [44] or capture of dynamic scenes [41, 43], 3D content generation [36, 52, 49], controllable 3D avatars [23], and prediction of 3D representations from few-shot images [5]. Methods that focus on more compact representation, thus suitable for rendering on mobile devices, have also been proposed. These methods prune/cluster Gaussians, adaptively selecting the number of spherical harmonics to encode color [32], and quantize the parameters of the representation [31]. All of these extensions are extremely recent, demonstrating the large interest sparked within the community. With the exception of [5], one core limitation that these methods share is that they all rely on the original *adaptive density control* heuristics that 3DGS [19] proposed. As we demonstrate in this work, this does not necessarily always work, and it require either careful initialization or appropriate tuning. Even then, the rendering outcomes may be suboptimal. For additional research, we direct interested readers to a recent survey [6].

**Concurrent work.** Bulo *et al.* [4] recently proposed to modify the densification strategy to address issues with cloning Gaussians, as well as densifying Gaussians at locations with high training error. Their method partially addresses the issue with cloning, but it is not enough as we will show in Section 3.4. Their error-based densification is orthogonal to our research direction and could easily be incorporated into our method as well, which we leave to future works. Other works explore how to better initialize through an auxiliary NeRF network [10] or via trained dense geometry estimators [9], such as [37]. In our case, we tackle the source of the problem and reduce the dependence on initialization itself.

## 3  Method

We first reformulate Gaussian Splatting as Markov Chain Monte Carlo (MCMC) sampling. We then introduce the new update equations under the MCMC framework with Stochastic Gradient Langevin Dynamics [3, 20]. We then discuss how the heuristics in 3D Gaussian Splatting [19] can be folded into a novel relocalization scheme. Finally, we discuss the L1 regularization that we use to encourage efficient use of the Gaussians, and the implementation details.

## 3.1 Brief review of 3D Gaussian Splatting

Before reformulating, we first briefly review 3D Gaussian Splatting [19] for completeness. 3D Gaussian Splatting represents the scene as a set of 3D Gaussians, which are then rasterized into a desired view via $\alpha$-blending. This can be viewed as an efficient way to perform volume rendering as in NeRFs [28]. Specifically, for a camera pose $\boldsymbol{\theta}$ to render a pixel $\mathbf{x}$ we order the $N$ Gaussians by sorting them in the order of increasing distance from the camera and write

$$\mathbf{C}(\mathbf{x}) = \sum_{i=1}^{N} \mathbf{c}_i \alpha_i(\mathbf{x}) \left[ \prod_{j=1}^{i-1} (1 - \alpha_j(\mathbf{x})) \right], \tag{1}$$

where $\mathbf{c}_i$ is the color of each Gaussian stored as Spherical Harmonics that are converted to color according to the pose $\boldsymbol{\theta}$, and if we denote their opacity as $o$, centers as $\boldsymbol{\mu}$, and covariance as $\boldsymbol{\Sigma}$,

$$\alpha_i(\mathbf{x}) = o_i \exp\left( -\tfrac{1}{2} (\mathbf{x} - \mathcal{R}(\boldsymbol{\mu}_i; \boldsymbol{\theta}))^{\mathrm{T}} \mathcal{R}_{\boldsymbol{\theta}} (\boldsymbol{\Sigma}_i)^{-1} (\mathbf{x} - \mathcal{R}(\boldsymbol{\mu}_i; \boldsymbol{\theta})) \right), \tag{2}$$

where $\mathcal{R}$ is the camera projection operation.

Then, with the color values for each pixel, Gaussians are trained to minimize the loss:

$$\mathcal{L}_{\text{orig}} = (1 - \lambda_{\text{D-SSIM}}) \cdot \mathcal{L}_1 + \lambda_{\text{D-SSIM}} \cdot \mathcal{L}_{\text{D-SSIM}}, \tag{3}$$

where $\mathcal{L}_1$ is the average L1 error between $\mathbf{C}(\mathbf{x})$ and the ground-truth colour $\mathbf{C}_{\text{gt}}(\mathbf{x})$, and $\mathcal{L}_{\text{D-SSIM}}$ is the Structural Similarity Index Metric (SSIM) [38] between the rendered and ground-truth image. Where $\lambda_{\text{D-SSIM}} = 0.2$ as proposed by [19].

## 3.2 3D Gaussian Splatting as Markov Chain Monte Carlo (MCMC)

Unlike existing approaches to 3D Gaussian Splatting, we propose to interpret the training process of placing and optimizing Gaussians as a *sampling* process. Rather than defining a loss function and simply taking steps towards a local minimum, we define a distribution $\mathcal{G}$ which assigns high probability to collections of Gaussians which faithfully reconstruct the training images. This choice allows us to leverage the power of MCMC frameworks to draw samples from this distribution in a way that is mathematically well-behaved, even when making discrete changes in the parameter space. As such, we can design discrete operations analogous to the original splitting and pruning heuristics of Gaussian Splatting without breaking the assumptions of continuity that underlie typical gradient-based optimization.

To achieve this, we start from the Stochastic Gradient Langevin Dynamics (SGLD) [40, 3] method, which is an MCMC framework that has also recently been applied to novel view synthesis applications [20]. This particular choice is convenient, as SGLD already resembles the commonly used Stochastic Gradient Descent (SGD) update rule, but with additional stochastic noise. Specifically, if we consider the updates of a single Gaussian $\mathbf{g}$ in 3DGS, and momentarily ignore their split/merge heuristics:

$$\mathbf{g} \leftarrow \mathbf{g} - \lambda_{\text{lr}} \cdot \nabla_{\mathbf{g}} \, \mathbb{E}_{\mathbf{I} \sim \mathcal{I}} \left[ \mathcal{L}_{\text{total}} (\mathbf{g}; \mathbf{I}) \right], \tag{4}$$

where $\lambda_{\text{lr}}$ is the learning rate, and $\mathbf{I}$ is an image sampled from the set of training images $\mathcal{I}$. Let us now compare the former to a typical SGLD update:

$$\mathbf{g} \leftarrow \mathbf{g} + a \cdot \nabla_{\mathbf{g}} \log \mathcal{P}(\mathbf{g}) + b \cdot \boldsymbol{\epsilon}, \tag{5}$$

where $\mathcal{P}$ is the data-dependent probability density function for the distribution one wishes to sample from, and $\boldsymbol{\epsilon}$ is the noise distribution for exploration. The hyperparameters $a$ and $b$ together control the trade-off between convergence speed and exploration[1] We note the striking similarity between (4) and (5). In other words, by having the loss as the negative log likelihood of the underlying distribution,

$$\mathcal{G} = \mathcal{P} \propto \exp(-\mathcal{L}_{\text{total}}), \tag{6}$$

the equations become identical if $\lambda_{\text{lr}} = -a$ and $b = 0$. Hence, the standard Gaussian Splatting optimization could be understood as having Gaussians that are sampled from a likelihood distribution that is tied to the rendering quality. We further note that this addition of noise to optimization is highly related to traditional optimization methods that inject noise [35, 30, 8] or that perform perturbed gradient descent [17, 18]. Here, we formulate it as MCMC in favour of the probabilistic relationship that it provides in (6), and the removal of heuristics that it enables, which we discuss in Sec. 3.4.

---

[1]In typical SGLD formalism, $b$ is typically expressed as a function of $a$ and an additional hyper-parameter, but we rewrite it in this form without any loss of generality following [20].

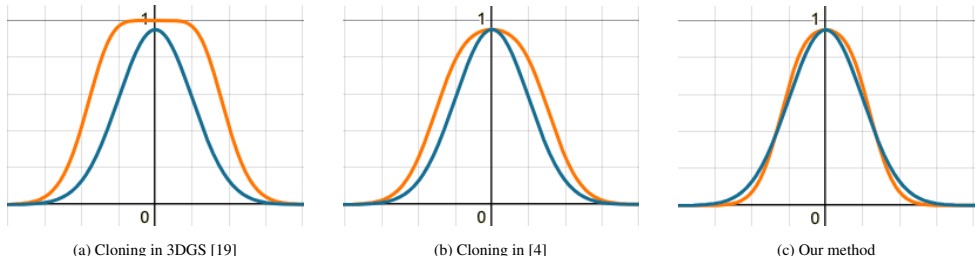

| (a) Cloning in 3DGS [19] | (b) Cloning in [4] | (c) Our method |

Figure 1: **Illustration of different respawn strategies –** We show a 1D example of rasterizing a Gaussian with opacity 0.95, **before** and **after** cloning them into four identical Gaussians and rasterizing them together, with different strategies. Existing methods cannot be used for MCMC as they broaden the extent of the selected Gaussian, significantly violating distribution invariance.

### 3.3 Updating with Stochastic Gradient Langevin Dynamics

Having revealed the link between SGLD and conventional 3DGS optimization, we rewrite (4) as:

$$\mathbf{g} \leftarrow \mathbf{g} - \lambda_{\text{lr}} \cdot \nabla_{\mathbf{g}} \mathbb{E}_{\mathbf{I} \sim \mathcal{I}} \left[ \mathcal{L}_{\text{total}} \left( \mathbf{g}; \mathbf{I} \right) \right] + \lambda_{\text{noise}} \cdot \boldsymbol{\epsilon}, \tag{7}$$

where $\lambda_{\text{lr}}$ and $\lambda_{\text{noise}}$ are the hyperparameters controlling the learning rate and the amount of exploration enforced by SGLD. In practice, instead of the raw gradients $\nabla_{\mathbf{g}} \mathbb{E}_{\mathbf{I} \sim \mathcal{I}} \left[ \mathcal{L}_{\text{total}} \left( \mathbf{g}; \mathbf{I} \right) \right]$, we use the Adam [21] optimizer with default parameters for $\beta_1$ and $\beta_2$ [25].

In Equation (7), it is important that the noise term $\boldsymbol{\epsilon}$ is designed carefully. The noise term $\boldsymbol{\epsilon}$ needs to be added in a way that it can be 'balanced' by the gradient term $\nabla_{\mathbf{g}_i} \mathcal{L}_{\text{total}}$, or otherwise (7) reduces to random updates. For example, it is quite common for Gaussians to be narrow when reconstructing scenes to represent lines and edges. Should the added noise force Gaussians to move to locations outside of the previous support regions, this random walk would be irrecoverable, breaking the MCMC sampling chain.

We further notice that exploration is not critical for opacity, scale, and color, and we *do not* add noise to these parameters. In fact, we empirically found that adding noise to them to slightly harm performance; see Section 4.1. Conversely, it has been shown by [11] that 3DGS reconstruction can be harmed significantly when areas of space are left unexplored, e.g. due to missing points in the initialization. This suggests potential for noise-based exploration to improve results for both random initialization as well as SFM-derived initializations which suffer from missing geometry.

Finally, as we are interested in the 'converged' quality not just exploration, we reduce the amount of noise when Gaussians are well-behaved, that is, when their opacities are high enough to be guided well by the gradients. Thus, we design the noise term *only on the locations of the Gaussians* such that it is dependent on their covariances and also its opacities, as well as the learning rate:

$$\boldsymbol{\epsilon}_\mu = \lambda_{\text{lr}} \cdot \sigma\left(-k(o - t)\right) \cdot \boldsymbol{\Sigma}\eta \quad \text{where} \quad \boldsymbol{\epsilon} = [\boldsymbol{\epsilon}_\mu, \mathbf{0}]. \tag{8}$$

where $\eta \sim \mathcal{N}(\mathbf{0}, \mathbf{I})$, $\sigma$ is the sigmoid function, and $k$ and $t$ are hyperparameters controlling the sharpness of the sigmoid, which we set as $k{=}100$ and $t{=}(1 - 0.005)$ to make a sharp transition function that goes from zero to one, centered around the default pruning threshold of 3D Gaussian Splatting [19] for the opacity values of Gaussians. In simple terms, (8) perturbs a Gaussian with anisotropic noise with the same anisotropy profile $\boldsymbol{\Sigma}$ of the Gaussian, while the sigmoid term reduces the effect of noise on opaque Gaussians.

### 3.4 Heuristics as state transitions via relocation

Inspired by jump and resampling moves in MCMC [26, 15], we now discuss how heuristics in 3D Gaussian Splatting can be rewritten as simple state transitions. In 3D Gaussian Splatting, heuristics are used to 'move', 'split', 'clone', 'prune', and 'add' Gaussians to encourage more 'live' Gaussians $(o_i{\geq}0.005)$.[2] We explain all of these modifications moving from one sample state $\mathbf{g}^{old}$ to another sample state $\mathbf{g}^{new}$. This applies also to cases where the number of Gaussians changes, as one could

---

[2]This is the default threshold used in 3D Gaussian Splatting [19].

think of the state with a smaller number of Gaussians simply as the equivalent state with more Gaussians, but with those that have zero opacity, that is, dead Gaussians. Importantly, for these kinds of deterministic moves to be integrated into MCMC frameworks, it is important that they do not cause MCMC sampling to collapse. Specifically, we aim to preserve the probability of the sample state before and after the move that is, $\mathcal{P}(\mathbf{g}^{new}) = \mathcal{P}(\mathbf{g}^{old})$ such that the move can be seen as simply hopping to another sample with equal probability. We now detail how we achieve this.

There can be various ways, but we opt for a simple strategy where we move 'dead' Gaussians ($o_i < 0.005$) to the location of 'live' Gaussians. In doing so, we set the parameters of the Gaussians to minimize the difference between the impact on renderings that $\mathbf{g}^{new}$ and $\mathbf{g}^{old}$ provide. We provide the exact derivation in Appendix A and here we provide the update equation. Without loss of generality, consider moving $N - 1$ Gaussians, $\mathbf{g}_{1,...,N-1}$, to $\mathbf{g}_N$. Then, denoting the old Gaussian parameters with superscript $old$ and the new ones with $new$, we write

$$\boldsymbol{\mu}_{1,...,N}^{new} = \boldsymbol{\mu}_N^{old}, \qquad o_{1,...,N}^{new} = 1 - \sqrt[N]{1 - o_N^{old}},$$

$$\boldsymbol{\Sigma}_{1,...,N}^{new} = \left(o_N^{old}\right)^2 \left( \sum_{i=1}^{N} \sum_{k=0}^{i-1} \left( \binom{i-1}{k} \frac{(-1)^k (o_N^{new})^{k+1}}{\sqrt{k+1}} \right) \right)^{-2} \boldsymbol{\Sigma}_N^{old}. \tag{9}$$

While, at first glance, the strategy we opt for may seem similar to 'cloning' in 3D Gaussian Splatting [19], the difference (9) brings is critical. In Figure 1, we illustrate this for a simplified 1D case, when $o_N^{old}=0.95$. As shown, classical cloning and the recently proposed 'centre corrected' version [4] both lead to a significant difference in the rasterized Gaussian as cloning is performed. This is because in (1), the composed opacity is a product of multiple Gaussian shapes and its negation. Both existing strategies lead to the extent of the selected Gaussian growing as shown in Fig. 1, thus significantly differ in terms of likeness of these states, that is $\mathcal{P}(\mathbf{g}^{new}) \neq \mathcal{P}(\mathbf{g}^{old})$. This, in fact, leads to sub-optimal training.

**Implementation.** While our method results in $\mathcal{P}(\mathbf{g}^{new}) \approx \mathcal{P}(\mathbf{g}^{old})$, it is not exact. Hence we apply this move every 100 iterations to avoid disruptions in the training process. To choose where to move, for each dead Gaussian, we first chose a target Gaussian to move/teleport to via multinomial sampling of the live Gaussians with the probabilities proportional to their opacity values. Please note that only *after* all movement decisions have been made we apply (9). Finally, as we rely on the Adam optimizer, moment statistics should also be adjusted. We reset the moment statistics for the target Gaussian (the original one that is cloned) so that it is biased to stay stationary, while for the new ones (source) we retain the moment statistics to encourage exploration. This is because 'dead' (source) Gaussians are dominated by the noise term in (7), and the moment statistics are hence appropriate to foster exploration.

### 3.5 Encouraging fewer Gaussians

To make effective use of the memory and compute while improving the performance, we encourage Gaussians to disappear in non-useful locations and 'respawn' elsewhere. As the existence of a Gaussian is effectively determined by its opacity $o$ and covariance $\boldsymbol{\Sigma}$, we apply regularization to both of these. Our full training loss is:

$$\mathcal{L}_{\text{total}} = (1 - \lambda_{\text{D-SSIM}}) \cdot \mathcal{L}_1 + \lambda_{\text{D-SSIM}} \cdot \mathcal{L}_{\text{D-SSIM}} + \lambda_o \cdot \sum_i |o_i|_1 + \lambda_{\boldsymbol{\Sigma}} \cdot \sum_{ij} \left| \sqrt{\text{eig}_j(\boldsymbol{\Sigma}_i)} \right|_1, \tag{10}$$

where $\text{eig}_j(.)$ denotes the $j$-th eigenvalues of the covariance matrix (the variance along the principle axes of the covariance matrix), and $\lambda_o$ and $\lambda_{\boldsymbol{\Sigma}}$ are hyperparameters.

### 3.6 More implementation details

We implement our method on top of the 3DGS [19] framework using PyTorch [33].

**Gradual increase in the number of Gaussians.** As in other works [19, 4], we allow the number of Gaussians to gradually grow, so that Gaussians are placed at useful locations. We do this simply by initially starting with a selected number of Gaussians, then allowing more 'dead' Gaussians to become 'alive' through our relocation strategy we previously detailed in Section 3.4. Specifically,

Table 1: **Quantitative results with same number of Gaussians –** Our method outperforms all baselines even when starting from random initialization, with a large gap in performance when compared with 3DGS [19] – Random. We highlight the **best** and *second-best* for each column.

| | NeRF Synthetic [28] | | | MipNeRF 360 [2] | | | Tank & Temples [22] | | | Deep Blending [16] | | | OMMO [27] | | |
|---|---|---|---|---|---|---|---|---|---|---|---|---|---|---|---|
| | PSNR↑ / SSIM↑ / LPIPS↓ | | | PSNR↑ / SSIM↑ / LPIPS↓ | | | PSNR↑ / SSIM↑ / LPIPS↓ | | | PSNR↑ / SSIM↑ / LPIPS↓ | | | PSNR↑ / SSIM↑ / LPIPS↓ | | |
| NeRF [29] | 31.01 / | - / | - | 24.85 / 0.66 / 0.43 | | | - | | | 21.18 / 0.78 / 0.34 | | | - | | |
| Plenoxels [46] | 31.76 / | - / | - | 23.63 / 0.67 / 0.44 | | | 21.08 / 0.72 / 0.38 | | | - | | | - | | |
| INGP-Big [29] | 33.18 / | - / | - | 26.75 / 0.75 / 0.30 | | | 21.92 / 0.75 / 0.31 | | | - | | | - | | |
| MipNeRF [1] | 33.09 / | - / | - | 27.60 / 0.81 / 0.25 | | | - | | | 21.54 / 0.78 / 0.37 | | | - | | |
| MipNeRF360 [2] | - | | | 29.23 / 0.84 / 0.21 | | | 22.22 / 0.76 / 0.26 | | | - | | | - | | |
| 3DGS [19]→ [4] | 33.32 / | - / | - | 28.69 / 0.87 / 0.22 | | | 23.14 / 0.84 / 0.21 | | | - | | | - | | |
| 3DGS [19] (Random) | *33.42* / **0.97** / **0.04** | | | 27.89 / 0.84 / 0.26 | | | 21.93 / 0.79 / 0.27 | | | 29.55 / **0.90** / 0.33 | | | 28.24 / 0.88 / 0.24 | | |
| Ours (Random) | **33.80** / **0.97** / **0.04** | | | *29.72* / *0.89* / **0.19** | | | *24.21* / **0.86** / **0.19** | | | **29.71** / **0.90** / **0.32** | | | *29.31* / *0.90* / **0.20** | | |
| 3DGS [19] (SfM) | - | | | 29.30 / 0.88 / 0.21 | | | 23.67 / 0.84 / 0.22 | | | 29.64 / **0.90** / **0.32** | | | 28.83 / 0.89 / 0.22 | | |
| Ours (SfM) | - | | | **29.89** / **0.90** / **0.19** | | | **24.29** / **0.86** / **0.19** | | | *29.67* / 0.89 / **0.32** | | | **29.52** / **0.91** / **0.20** | | |

we gradually increase the number of live Gaussians by 5% until the maximum desired number of Gaussians is met.

**Initialization and training.** We initialize our samples either randomly or from point clouds, typically from Structure-from-Motion (SfM) as in 3DGS [19]. For random initialization, we follow 3DGS [19] and uniformly random sample $100k$ Gaussians within three times the extent of the camera bounding box. We also use the same learning rate and learning-rate schedulers to enable comparisons. For the location of Gaussians, we start at a learning rate of $1.6e^{-4}$ and decay it exponentially to $1.6e^{-6}$. For all experiments, unless specified otherwise, we use $\lambda_{\text{noise}}=5 \times 10^5$, $\lambda_{\Sigma}=0.01$, and $\lambda_o=0.01$. For Deep Blending [16], we use $\lambda_o=0.001$. Following 3DGS [19], we start with 500 warmup iterations, during which we do not perform our relocalization in Sec. 3.4 nor increase the number of Gaussians.

## 4  Experiments

We use various datasets, both synthetic and real. Specifically, as in 3DGS [19], we use all scenes from NeRF Synthetic [28] dataset, the same two scenes used in [19] of Tank & Temples [22], and Deep Blending [16] and all publicly available scenes from MipNeRF 360 [2]. We do not use 'Flower' and 'Treehill' scenes from MipNeRF 360 [2] as they are not publicly available. We further use all scenes from the OMMO [27] dataset as in [10], for the large scenes with distant objects it provides. For MipNeRF 360 [2], to make our results compatible with [19], we downsample the indoor scenes by a factor of two, and the outdoor scenes four. For OMMO [27] scene #01, we downsample the images four times to keep the image size reasonable ($1000 \times 750$). For all other scenes, we use the original image resolutions. In the main paper, we report our results by summarizing the average statistics for each dataset. Results for individual scenes, including the standard deviation of multiple runs, can be found in Appendix B. License information for each dataset can be found in Appendix E.

**Metrics.** We evaluate each method using three standard metrics: Peak Signal-to-Noise Ratio (PSNR), Structural Similarity Index Metric (SSIM) [38], and Learned Perceptual Image Patch Similarity (LPIPS) [50]. To account for randomness, we run all experiments three times and average the results.

**Baselines.** We compare against conventional 3DGS [19], with both random and SfM point cloud-based initialization strategies. We use their official code for our experiments. We also report the original numbers in 3DGS [19], but where we correct their LPIPS scores, as reported by [4]. We also include state-of-the-art baselines for each dataset.

### 4.1  Results

**Performance with the same number of Gaussians.** As the number of Gaussians is directly related to the quality of novel-view rendering, we first compare our method with existing baselines using the *same* number of Gaussians as 3DGS [19]. In more details, we simply set the number of Gaussians used in the original 3DGS [19] and set it as our maximum number of Gaussians to be used during training and inference. We show the quantitative results in Table 1, and qualitative highlights

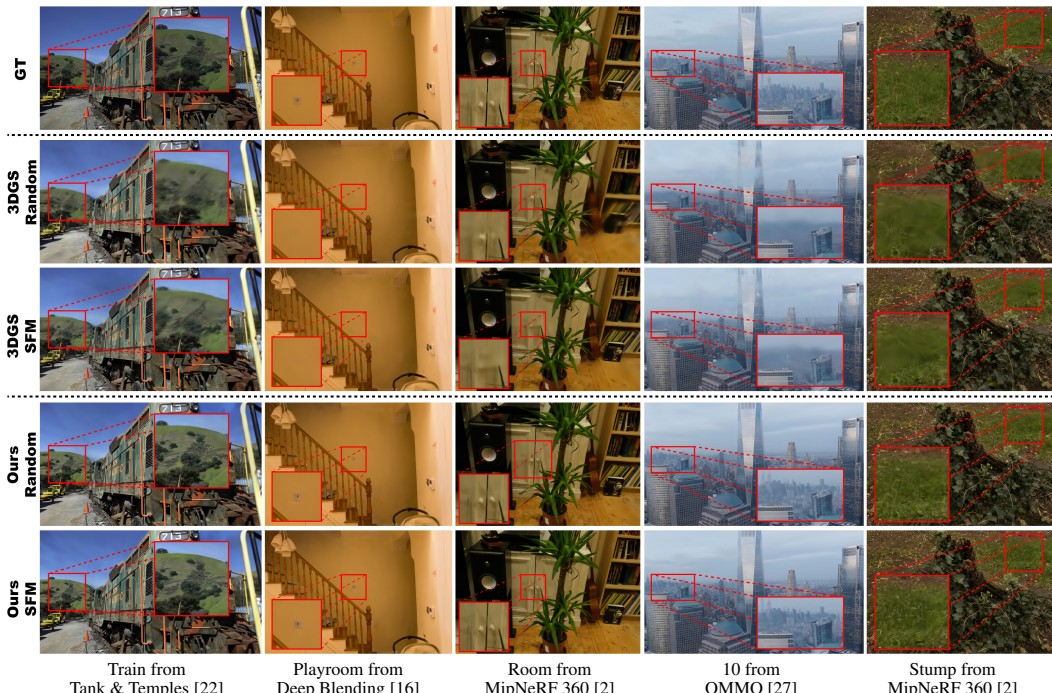

| | Train from Tank & Temples [22] | Playroom from Deep Blending [16] | Room from MipNeRF 360 [2] | 10 from OMMO [27] | Stump from MipNeRF 360 [2] |

Figure 2: **Qualitative highlights with the same number of Gaussians –** We provide examples of novel-view rendering of 3DGS [19] and our approach on multiple scenes from different datasets (with either random or SFM initialization). We highlight the differences in inset figures. Our method faithfully represents details of the various regions thanks to our hybrid MCMC re-formulation that allows exploration without heuristics. Our results provide higher quality reconstructions. Please zoom-in to see details.

in Figure 2. As shown, our method provides better performance than 3DGS [19]. Interestingly, our method, whether initialized randomly or with SfM point clouds, only displays minor variations in performance, thanks to the exploration that our MCMC formulation provides. This allows our method to outperform 3DGS [19], and *regardless* of the initialization strategy.

**Limited budget.** We further verify the effectiveness of our formulation by limiting the budget for the number of Gaussians on all the datasets (we omit NeRF Synthetic [28], as performance on the synthetic dataset is highly saturated, as also highlighted by 3DGS [19] in their initialization experiments). To limit the number of Gaussians for the conventional 3DGS [19], we simply stop their densification strategy from spawning more points if the limit on the number of Gaussians has been reached. Note that the pruning strategy can cause densification to resume, should some Gaussians get pruned after this threshold is met. Note also that our experiments for this setup is using 3DGS [19] with its default parameters, and do not perform further hyper-tuning. We use the *exact same* hyperparameters as in Sec. 3.6 for this experiment as well. We report the summary of the results in Fig. 3. With a limited budget, the gap between our method and 3DGS [19] increases.

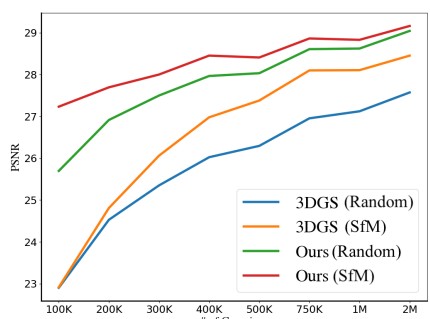

Figure 3: **Varying the #Gaussians –** We report the PSNR of 3DGS [19] and our method averaged over all datasets (except NeRF Synthetic).

**Sensitivity to initialization.** An important benefit of our method is that it allows exploration through the MCMC sampling scheme, removing the heavy reliance of 3DGS [19] on initialization. To verify this, we deviate from the default initialization strategy of 3DGS [19], which is to randomly place Gaussians within $3\times$ the camera extent as defined by [19]. Instead, we use $1\times$ the camera extent.

Table 3: **Ablation study –** We report the quantitative metrics without various components of our method, and with the L1 regularizer in 3D Gaussian Splatting [19]. We use the MipNeRF 360 [2] dataset, and random initialization. All components contribute to the final rendering quality.

| 3DGS [19]
PSNR↑ / SSIM↑ / LPIPS↓ | 3DGS [19]
w/ $\mathcal{L}_{\text{total}}$
PSNR↑ / SSIM↑ / LPIPS↓ | Ours
w/ $\mathcal{L}_{\text{orig}}$
PSNR↑ / SSIM↑ / LPIPS↓ | Ours
$\lambda_{\text{noise}}$=0
PSNR↑ / SSIM↑ / LPIPS↓ | Ours
Noise on all param.
PSNR↑ / SSIM↑ / LPIPS↓ | Ours
Full Method
PSNR↑ / SSIM↑ / LPIPS↓ |
|---|---|---|---|---|---|
| 27.89 / 0.84 / 0.26 | 23.84 / 0.77 / 0.33 | 23.90 / 0.67 / 0.42 | 27.41 / 0.83 / 0.26 | 29.11 / 0.86 / 0.24 | **29.72 / 0.89 / 0.19** |

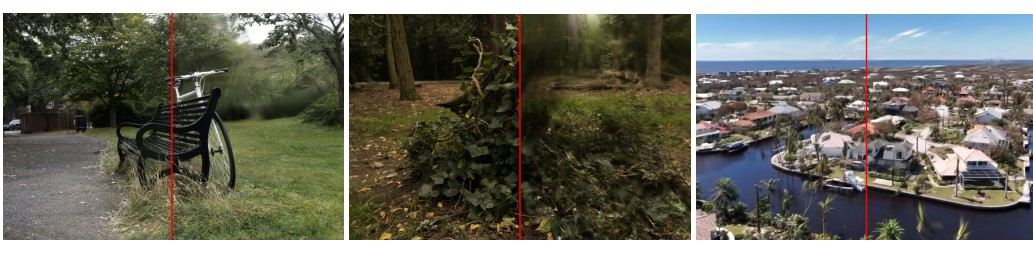

(a) 'Bicycle' from MipNeRF 360 [2]    (b) 'Stump' from MipNeRF 360 [2]    (c) '03' from OMMO [27]

Figure 4: **Effect of the noise term ($\epsilon$) –** We visualize our reconstruction with (left half) and without (right half) the noise term in (7). The noise terms are essential to explore the full scene extent.

We summarize our results for the average of all scenes in MipNeRF 360 [2] in Table 2. As shown, our method provides robustness to the initialization strategy, whereas 3DGS [19] requires careful initialization. This is also evident in Section 4.1, where our results with random initialization remain competitive with those initialized by SfM point clouds.

Table 2: **Initialization ablation –** Our method provides a similar performance regardless of the initialization strategy, whereas the performance of the original 3DGS [19] differs significantly.

|  |  | 3× camera extent [19]
PSNR↑ / SSIM↑ / LPIPS↓ | 1× camera extent
PSNR↑ / SSIM↑ / LPIPS↓ |
|---|---|---|---|
| 3DGS [19] | (Random) | 27.89 / 0.84 / 0.26 | 22.72 / 0.75 / 0.34 |
| Ours | (Random) | **29.72 / 0.89 / 0.19** | **29.64 / 0.89 / 0.19** |

**Ablations.** We further perform ablation studies in Tab. 3. We evaluate whether our regularizers defined on opacity and scale of Gaussians can help conventional 3DGS [19] as well and how essential they are for our method, using the MipNeRF 360 [2] dataset and with random initialization, as it makes their effect easier to observe. In the case of our method, where exploration is encouraged, they are essential to prevent stray Gaussians that shoot off into spaces that are not well updated by the reconstruction loss, *e.g.*, regions outside of the view frustum. The existence of noise is critical to achieving best performance, as without it Gaussians cannot explore the full extent of the scene, as we also illustrate in Fig. 4. Further, our regularizers are rather harmful when used with classical 3DGS [19], as they are not compatible with the heuristics proposed therein.

We also explore in Tab. 3 adding noise to other parameters. In addition to our location noise in (8), we apply $\mathcal{N}(\mathbf{0}, \mathbf{I})$ to scale and rotation and $\mathcal{N}(\mathbf{0}, 0.1 \times \mathbf{I})$ for opacity, and decrease them exponentially with a decay rate of 0.9995 for each iteration. This slightly worse performance suggests that the other parameters do not require as much exploration as the Gaussian locations do.

On Tank & Temples [22], we additionally investigate the design choice of our noise scheduler by removing the covariance term and the opacity term. Without the covariance term, we achieve a PSNR of 23.16, whereas with the covariance term 24.21. Without the opacity term, we achieve 22.47, which is also performing worse than with the two terms. We note that removing the opacity term requires $\lambda_{\text{noise}}$ to be set to 0.05 or the model fails to train (PSNR < 7).

Finally, we also compare the impact that noise scheduling has. Using the same exponential scheduling as in 3DGS [19]—which is what we use—achieves best performance of 24.21 PSNR, while a linear scheduler gives 17.64. We further compare with the scheduler suggested in [30], which gives 22.46.

## 4.2 Computational time

As our method results in the same 3D Gaussian Splat representation as prior work, inference time is the same with 3DGS [19], which is highly efficient. To provide an exact comparison, we took our Gaussians trained at 1M Gaussian, and re-measured the single optimization iteration time for

Table 4: **Timings –** We report the timings of our method with various opacity regularizer ($\lambda_o$) settings and the maximum number of Gaussians to 300k. Our method can deliver faster training while still maintaining reconstruction quality.

|      | 3DGS [19]  | Ours ($\lambda_o$=0.01) | Ours ($\lambda_o$=0.001) | Ours ($\lambda_o$=0.01, 300k) |
|------|-----------|-------------------------|--------------------------|-------------------------------|
| PSNR | 31.7      | 32.5                    | 32.4                     | 31.8                          |
| Time | 25 minutes | 42 minutes             | 30 minutes               | 21 minutes                    |

our method and the original 3DGS. In this case, ours takes 80 milliseconds while 3DGS takes 76 milliseconds. That is, the added time for sampling and noise addition is not substantial, even with our implementation that implements resampling naively with PyTorch [33]'s `torch.multinomial`—this could be further accelerated with a CUDA implementation.

Furthermore, the configuration of Gaussians (i.e. where they are, their sizes and opacity) matters greatly to the runtime because they affect the speed of rasterization (among them the opacity regularizer affects the speed the greatest). Hence, we evaluate the runtime for the 'Room' scene in MipNeRF 360 dataset, all with SfM initializations and maximum number of 1.5M Gaussians per the original 3DGS implementation. We report the timings in Tab. 4. As shown, our method, while it may take longer to achieve the highest PSNR, trains faster when a similar PSNR is desired. An independent implementation of our method [45] also confirms a *20% reduction in training time* and a *65% reduction in required memory* when using our method.

## 5 Conclusion

In this paper, we reformulated 3D Gaussian Splatting [19] training as Markov Chain Monte Carlo (MCMC) and implement it via Stochastic Gradient Langevin Dynamics (SGLD). By doing so, we show that we can eliminate the need for point-cloud initialization, and avoid heuristic-based densification, pruning and reset. Not only do we show that this strategy generalizes well across various scenes, outperforming the original 3D Gaussian Splatting [19], but for the *first time* we show that this leads to a 3DGS implementation that beats NeRF backbones on the challenging MipNeRF3360 [2] dataset.

**Acknowledgements** We would like to thank Charatan et al. [5], as the inspiration to deal with 3DGS optimization as a distribution was inspired by their work. Similarly, we would like to thank Goli et al. [14], as the notion of adding noise to Gaussians stemmed from wanting to explore their positional null-space. We would also like to thank Sergio Orts Escolano and Federico Tombari for their feedback. This work was supported in part by the Natural Sciences and Engineering Research Council of Canada (NSERC) Discovery Grant, NSERC Collaborative Research and Development Grant, Google, Digital Research Alliance of Canada, and Advanced Research Computing at the University of British Columbia, and by the SFU Visual Computing Research Chair program.

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

# Appendix

## A  Derivation for the cloning strategy

To have minimal impact on the rendering outcome we propose a strategy that minimizes the difference between the rasterization outcomes of a Gaussian, before and after cloning. For simplicity, let us drop the subscript for the Gaussian index, and consider only the case when a selected Gaussian is to be cloned to multiple copies. We set $\mathbf{c}^{new}{=}\mathbf{c}^{old}$ for the cloned Gaussians, and also, as the Gaussians we use in 3D Gaussian Splatting are unnormalized, we set their central $\mathbf{C}(\mathbf{x})$ values to be identical before and after the split, that is, $\mathbf{C}^{new}(\boldsymbol{\mu}){=}\mathbf{C}^{old}(\boldsymbol{\mu})$. Plugging $\mathbf{x} = \boldsymbol{\mu}$ in (1), this means

$$(1 - o^{new})^N = 1 - o^{old}, \tag{11}$$

which is the same as [4] when $N{=}2$.

However, (11) is not enough to preserve rasterization as already noted in [4], for these Gaussians are not point sources—we thus need to alter the covariance $\boldsymbol{\Sigma}$ of these Gaussians. One way to guarantee minimal impact would be to minimize the mean squared error, that is

$$\text{minimize} \int_{-\infty}^{\infty} \left\| \mathbf{C}^{new}(\mathbf{x}) - \mathbf{C}^{old}(\mathbf{x}) \right\|_2^2 d\mathbf{x}, \tag{12}$$

But solving this results in a complex equation without a simple analytical form.

We thus opt for an alternative solution, where we consider random 1D slices of Gaussians that pass through their centers, inspired by sliced Wasserstein methods [24]. Given that our Gaussians share their centers, the integral of the rasterization before and after the cloning operation *must* be equal for rasterization to remain the same for *any* arbitrary slice. Thus, denoting points along this arbitrary slice $\mathcal{S}(\cdot)$ as $x = \mathcal{S}(\mathbf{x})$, we instead write

$$\text{minimize} \left\| \int_{-\infty}^{\infty} \mathbf{C}^{new}(x)dx - \int_{-\infty}^{\infty} \mathbf{C}^{old}(x)dx \right\|_2^2, \quad \forall \mathcal{S}. \tag{13}$$

This equation has a simple analytical solution and we can achieve $\int_{-\infty}^{\infty} \mathbf{C}^{new}(x)dx = \int_{-\infty}^{\infty} \mathbf{C}^{old}(x)dx$ in many ways, including our update equation in (9). Given this, we now derive how we modify the scales.

Without loss of generality, let us consider the mean, $\boldsymbol{\mu}$, of the cloned Gaussian to be zero to simplify the equations further. In addition to the points along the slice $x$, let us further denote the 'sliced' covariance as $\Sigma = \mathcal{S}(\boldsymbol{\Sigma})$. Then, by plugging (2) into (1), for the new Gaussians, we can now write

$$
\begin{aligned}
\mathbf{C}^{new}(x) &= \sum_{i=1}^{N} o^{new} \exp\left(-\frac{x^2}{2\Sigma^{new}}\right) \prod_{j=1}^{i-1} \left(1 - o^{new} \exp\left(-\frac{x^2}{2\Sigma^{new}}\right)\right), \\
&= \sum_{i=1}^{N} o^{new} \exp\left(-\frac{x^2}{2\Sigma^{new}}\right) \left(1 - o^{new} \exp\left(-\frac{x^2}{2\Sigma^{new}}\right)\right)^{i-1}.
\end{aligned}
\tag{14}
$$

Interestingly, the $N$-power here can be made even simpler by the fact that for $p < 1$

$$(1 - p)^N = \sum_{k=0}^{N} \binom{N}{k} (-1)^k (p)^k, \tag{15}$$

which allows us to rewrite (14) as

$$
\begin{aligned}
\mathbf{C}^{new}(x) &= \sum_{i=1}^{N} \sum_{k=0}^{i-1} \binom{i-1}{k} (-1)^k (o^{new})^{k+1} \exp\left(-\frac{x^2}{2\Sigma^{new}}\right)^{(k+1)}, \\
&= \sum_{i=1}^{N} \sum_{k=0}^{i-1} \binom{i-1}{k} (-1)^k (o^{new})^{k+1} \exp\left(-\frac{(k+1)x^2}{2\Sigma^{new}}\right).
\end{aligned}
\tag{16}
$$

Table 5: **All results with same number of Gaussians –** We report the performance of our method and 3DGS [19] with the same number of Gaussians. Tab. 1 reports the average entries from this table. Our method outperforms 3DGS [19] across the board. We do not include results for non-random initialization on NeRF Synthetic as this dataset does not include COLMAP point clouds.

| | | 3DGS [19] (Random) | 3DGS [19] (SfM) | Ours (Random) | Ours (SfM) |
|---|---|---|---|---|---|
| | | PSNR↑ / SSIM↑ / LPIPS↓ | PSNR↑ / SSIM↑ / LPIPS↓ | PSNR↑ / SSIM↑ / LPIPS↓ | PSNR↑ / SSIM↑ / LPIPS↓ |
| NeRF Synthetic [28] | Mic | 35.35 / 0.99 / 0.01 | – | 37.29 / 0.99 / 0.01 | – |
| | Ship | 30.93 / 0.90 / 0.13 | – | 30.82 / 0.91 / 0.12 | – |
| | Lego | 35.84 / 0.98 / 0.02 | – | 36.01 / 0.98 / 0.02 | – |
| | Chair | 36.17 / 0.99 / 0.02 | – | 36.51 / 0.99 / 0.02 | – |
| | Materials | 30.00 / 0.96 / 0.04 | – | 30.59 / 0.96 / 0.04 | – |
| | Hotdog | 37.83 / 0.99 / 0.03 | – | 37.82 / 0.99 / 0.02 | – |
| | Drums | 26.22 / 0.95 / 0.04 | – | 26.29 / 0.95 / 0.04 | – |
| | Ficus | 35.01 / 0.99 / 0.01 | – | 35.07 / 0.99 / 0.01 | – |
| | Average | 33.42 / 0.97 / 0.04 | – | 33.80 / 0.97 / 0.04 | – |
| | Std | 0.0076 / 0.0000 / 0.0000 | – | 0.0105 / 0.0000 / 0.0000 | – |
| MipNeRF 360 [2] | Counter | 28.09 / 0.88 / 0.30 | 29.12 / 0.91 / 0.24 | 29.16 / 0.92 / 0.23 | 29.51 / 0.92 / 0.22 |
| | Stump | 23.91 / 0.68 / 0.33 | 26.99 / 0.78 / 0.24 | 27.67 / 0.82 / 0.20 | 27.80 / 0.82 / 0.19 |
| | Kitchen | 30.54 / 0.92 / 0.16 | 31.58 / 0.93 / 0.14 | 32.23 / 0.94 / 0.14 | 32.27 / 0.94 / 0.14 |
| | Bicycle | 24.57 / 0.70 / 0.33 | 25.64 / 0.78 / 0.23 | 26.06 / 0.81 / 0.19 | 26.15 / 0.81 / 0.18 |
| | Bonsai | 30.94 / 0.93 / 0.26 | 32.32 / 0.95 / 0.24 | 32.67 / 0.95 / 0.23 | 32.88 / 0.95 / 0.22 |
| | Room | 30.04 / 0.90 / 0.32 | 31.70 / 0.93 / 0.27 | 32.30 / 0.94 / 0.25 | 32.48 / 0.94 / 0.25 |
| | Garden | 27.16 / 0.86 / 0.14 | 27.73 / 0.87 / 0.12 | 27.99 / 0.88 / 0.11 | 28.16 / 0.89 / 0.10 |
| | Average | 27.89 / 0.84 / 0.26 | 29.30 / 0.88 / 0.21 | 29.72 / 0.89 / 0.19 | 29.89 / 0.90 / 0.19 |
| | Std | 0.0524 / 0.0021 / 0.0016 | 0.0276 / 0.0003 / 0.0003 | 0.0246 / 0.0001 / 0.0003 | 0.0154 / 0.0001 / 0.0001 |
| Tank & Temples [22] | Train | 21.24 / 0.77 / 0.30 | 21.94 / 0.81 / 0.25 | 22.40 / 0.83 / 0.24 | 22.47 / 0.83 / 0.24 |
| | Truck | 22.63 / 0.82 / 0.24 | 25.40 / 0.88 / 0.18 | 26.02 / 0.89 / 0.14 | 26.11 / 0.89 / 0.14 |
| | Average | 21.93 / 0.80 / 0.27 | 23.67 / 0.84 / 0.22 | 24.21 / 0.86 / 0.19 | 24.29 / 0.86 / 0.19 |
| | Std | 0.0521 / 0.0008 / 0.0007 | 0.0639 / 0.0004 / 0.0004 | 0.0729 / 0.0007 / 0.0007 | 0.0688 / 0.0004 / 0.0008 |
| Deep Blending [16] | Dr Johnson | 28.94 / 0.89 / 0.34 | 29.14 / 0.90 / 0.33 | 29.05 / 0.89 / 0.33 | 29.00 / 0.89 / 0.33 |
| | Playroom | 30.16 / 0.90 / 0.32 | 30.15 / 0.90 / 0.32 | 30.37 / 0.90 / 0.31 | 30.33 / 0.90 / 0.31 |
| | Average | 29.55 / 0.90 / 0.33 | 29.64 / 0.90 / 0.32 | 29.71 / 0.90 / 0.32 | 29.67 / 0.89 / 0.32 |
| | Std | 0.0688 / 0.0007 / 0.0010 | 0.0487 / 0.0003 / 0.0002 | 0.0556 / 0.0015 / 0.0010 | 0.0458 / 0.0022 / 0.0022 |
| OMMO [27] | 01 | 25.13 / 0.77 / 0.27 | 25.64 / 0.79 / 0.25 | 25.90 / 0.80 / 0.23 | 25.89 / 0.80 / 0.22 |
| | 03 | 25.61 / 0.86 / 0.27 | 25.80 / 0.87 / 0.26 | 27.38 / 0.89 / 0.22 | 27.62 / 0.89 / 0.22 |
| | 05 | 27.66 / 0.86 / 0.29 | 28.37 / 0.87 / 0.28 | 28.81 / 0.88 / 0.27 | 28.87 / 0.88 / 0.27 |
| | 06 | 27.12 / 0.91 / 0.25 | 26.79 / 0.91 / 0.25 | 27.52 / 0.94 / 0.20 | 27.65 / 0.94 / 0.19 |
| | 10 | 29.64 / 0.87 / 0.26 | 29.99 / 0.89 / 0.23 | 31.20 / 0.90 / 0.22 | 31.51 / 0.91 / 0.20 |
| | 13 | 31.60 / 0.92 / 0.22 | 32.75 / 0.94 / 0.18 | 32.65 / 0.94 / 0.18 | 33.14 / 0.95 / 0.16 |
| | 14 | 30.33 / 0.93 / 0.19 | 30.87 / 0.94 / 0.17 | 31.03 / 0.94 / 0.17 | 31.26 / 0.94 / 0.16 |
| | 15 | 28.79 / 0.90 / 0.19 | 30.46 / 0.93 / 0.16 | 29.96 / 0.93 / 0.16 | 30.25 / 0.93 / 0.15 |
| | Average | 28.24 / 0.88 / 0.24 | 28.83 / 0.89 / 0.22 | 29.31 / 0.90 / 0.20 | 29.52 / 0.91 / 0.20 |
| | Std | 0.0265 / 0.0006 / 0.0007 | 0.0299 / 0.0004 / 0.0003 | 0.0233 / 0.0002 / 0.0001 | 0.0226 / 0.0001 / 0.0003 |

Finally, as $\int_{-\infty}^{\infty} \exp(-ax^2) = \sqrt{\pi/a}$ for some value $a$, plugging (16) into $\int_{-\infty}^{\infty} \mathbf{C}^{new}(x)dx = \int_{-\infty}^{\infty} \mathbf{C}^{old}(x)dx$ we write

$$\sum_{i=1}^{N}\sum_{k=0}^{i-1}\binom{i-1}{k}(-1)^k(o^{new})^{k+1}\sqrt{\frac{2\pi\Sigma^{new}}{k+1}} = o^{old}\sqrt{2\pi\Sigma^{old}}. \tag{17}$$

Solving for $\Sigma^{new}$ we get

$$\Sigma^{new} = \left(o^{old}\right)^2\left(\sum_{i=1}^{N}\sum_{k=0}^{i-1}\binom{i-1}{k}\frac{(-1)^k(o^{new})^{k+1}}{\sqrt{k+1}}\right)^{-2}\Sigma^{old}. \tag{18}$$

Since we want (18) to hold for any arbitrary slice $\mathcal{S}(\cdot)$, we can simply replace $\Sigma$ with $\mathbf{\Sigma}$, which is then the update equation in (9).

## B Detailed results

We report all numbers for all scenes in Tab. 5. The standard deviation (std) is computed based on the averages obtained from three different runs, where each average is calculated across all scenes in a dataset for a given seed.

## C Limitations and future work

While our method allows more robustness to initialization and higher rendering quality thanks to the exploration introduced by MCMC, it is still subject to the same limitations as 3DGS [19] in terms of its modelling capacity. For example, the aliasing issue solved in [48] can also be a problem with our method, as well as modelling reflections [44]. Our method, however, should be compatible with these advancements in Gaussian Splatting, as our method can be viewed as a better training framework for Gaussian Splats. In other words, it should enhance all other Gaussian Splatting methods that are available.

## D Broader impact

While our method is focusing on the core problem of 3D reconstruction, thus not having immediate societal implications, it may, however, have a rippling impact on downstream applications. Because our method reduces the reliance that Gaussian Splatting has on initialization, it may enhance these downstream applications. For example, 3D content generation [36, 52, 49] often suffers from the heuristics in Gaussian Splatting, which we remove. For controllable human modeling [23], our method also has the potential to enhance its quality. Thus, our method may indirectly enhance what is capable with generative methods, as well as human avatars. Both applications have the potential to be misused. This is a concern with any technology, and we urge users to think about the implications before applying our method.

## E Dataset licenses

We use the following datasets:

- NeRF Synthetic [28]: made available under Creative Commons Attribution 3.0 License. Available at `https://www.matthewtancik.com/nerf`.
- Mip-NeRF 360 [2]: no license terms provided. Available at `https://jonbarron.info/mipnerf360/`.
- OMMO [27]: no license terms provided. Available at `https://ommo.luchongshan.com/`.
- Deep Blending [16]: no license terms provided. Available at `http://visual.cs.ucl.ac.uk/pubs/deepblending/`.
- Tank & Temples [22]: made available under Creative Commons Attribution-NonCommercial-ShareAlike 3.0 License `https://www.tanksandtemples.org/license/`.

