# OpenReview forum: "3D Gaussian Splatting as Markov Chain Monte Carlo"
_NeurIPS.cc/2024/Conference — NeurIPS 2024 spotlight_

### Official Review · Reviewer_cJQR · 2024-07-13

**Soundness:** 3
**Presentation:** 3
**Contribution:** 3
**Rating:** 5
**Confidence:** 4

**Summary:**

This paper proposes a novel densification strategy of 3D Gaussian Splatting (3DGS) based on the Markov Chain Monte Carlo (MCMC) sampling scheme.
The authors address the ‘heuristic’ densification of standard 3DGS and adopt a distribution-aware resampling pipeline.
Consequently, they achieve higher rendering quality compared to standard 3DGS with a similar number of Gaussian primitives.

**Strengths:**

This paper is well-written to understand.

- The authors tackle the limitation of heuristic densification of standard 3DGS and suggest an MCMC sampling strategy for 3DGS. Also, the detailed analysis effectively supports their theoretical statements.

- It achieves standard 3DGS in rendering quality while preserving the fast inference time with the same format of 3DGS representation. Therefore, it can be used in various applications of 3DGS without any modification.

**Weaknesses:**

- The additional computational costs are required for MCMC resulting in the increase of training time compared to standard 3DGS.

- They only suggest the resampling strategy of 3DGS. Therefore, the technical contribution is inefficient despite the detailed analysis.

**Questions:**

- Although they have mentioned the computational time in Sec. B of the appendix, I cannot understand how much it takes compared to standard 3DGS. Can you provide the comparison in training time of this method and standard 3DGS for each scene?

- Although it has been described in L184-186, I am confused about the reason why adding noise to other parameters leads to harmful results. Can you describe more details about it?

**Limitations:**

It only tackles the cloning strategy without any other problems for achieving higher rendering quality. Thus, the technical improvement seems to be inefficient.

---

> ### Author Rebuttal · Authors · 2024-08-06
>
> We thank the reviewer for their constructive comments and suggestions.
> Here are our responses to your comments:
>
> ## Technical contribution
>
> We respectfully disagree, as our method cannot be summarized as a mere resampling strategy. It is a completely new take on the 3D Gaussian Splatting optimization that removes ALL the non-differentiable heuristics (and inherent training instability) that were introduced by the original authors: “opacity reset”, “densification”, "splitting", and “pruning”.
> We have further shown that this is non-trivial to achieve, as one must carefully design components in a way that obeys the basic principles of hybrid MCMC. Thanks to our re-formulation, future works could integrate various advancements in perturbed Gradient Descent, as reviewer Cpa7 suggests.
>
> ## Additional compute cost
>
> Please see the global response for the training time. In short, considering only the compute time for each Gaussian the added time is negligible. Ours does run slower if trained with a high opacity regularizer due to more transparent Gaussians, but it ultimately gives higher quality results, and it also can be tuned to **train faster while still outperforming 3DGS**.
>
>
> ## Noise to other parameters being harmful
>
> There have been a number of recent papers demonstrating that the locations of the Gaussians suffer heavily from local optima, which leads to their reliance on good initializations [7, 8]. Hence, adding noise to these parameters helps escape these local optima, and Gaussians to “explore” the scene. Other degrees of freedom (rotation, scale, opacity) do not suffer as heavily, hence the noise simply slows down convergence unnecessarily. Please see additional details showing more experiments on the noise design in our answer to reviewer Cpa7 if the reviewer is interested.

---

> > ### Comment · Reviewer_cJQR · 2024-08-13
> >
> > Thank you for the responses. The authors have addressed all of my concerns in the rebuttal. Thus, I have decided to maintain my rating in support of acceptance.

---

### Official Review · Reviewer_m7Kb · 2024-07-13

**Soundness:** 3
**Presentation:** 3
**Contribution:** 3
**Rating:** 7
**Confidence:** 2

**Summary:**

The paper discusses improvements to 3D Gaussian Splatting in neural rendering. Current methods rely on complex cloning and splitting strategies for placing Gaussians, which often do not generalize well and depend heavily on good initializations. The authors propose rethinking 3D Gaussians as random samples from an underlying probability distribution of the scene, using Markov Chain Monte Carlo (MCMC) sampling. They show that 3D Gaussian updates can be converted into Stochastic Gradient Langevin Dynamics (SGLD) updates by adding noise. This allows for the removal of heuristic densification and pruning strategies, replacing them with a deterministic state transition of MCMC samples. Additionally, the authors introduce an L1-regularizer on Gaussians to encourage efficient usage. Their method improves rendering quality, provides easy control over the number of Gaussians, and is robust to initialization across various standard evaluation scenes.

**Strengths:**

1) This work is very insightful
2) Experiments are extensive and detailed

**Weaknesses:**

/

**Questions:**

/

**Limitations:**

/

---

> ### Author Rebuttal · Authors · 2024-08-06
>
> We thank the reviewer for the positive assessment of our paper. Please let us know if you have any concerns or questions and we will try our best to respond.

---

> > ### Comment · Reviewer_m7Kb · 2024-08-14
> >
> > Thanks for the authors' sincere effort and detailed experiments. I keep my original score.

---

### Official Review · Reviewer_jFyG · 2024-07-13

**Soundness:** 4
**Presentation:** 4
**Contribution:** 4
**Rating:** 8
**Confidence:** 5

**Summary:**

Current 3DGS-based methods require carefully designed strategies such as cloning and splitting to assign a 3D Gaussian at a location. Further, they also require initializing points from SFM to generate high-quality novel views. The proposed work assumes that a set of 3D Gaussians are drawn from an underlying probability distribution, which is representative of the scene. Further, 3DGS updates are converted to SGLD updates by introducing noise.

The main contributions of this work are as follows:
- A fresh perspective that 3D Gaussians are sampled from a distribution and relocation strategy is compatible with MCMC samples.
- Robustness to initialization. 3DGS-MCMC is not dependent on the initialization step in 3DGS.
- The proposed method outperforms other NeRF-based methods and 3DGS on standard datasets.

**Strengths:**

- **Qualitative and Quantitative Results:** The proposed method is evaluated on NeRF synthetic, Tank&temples, Deep Blending, MipNeRF360 and OMMO dataset. Quantitatively, 3DGS-MCMC outperforms NeRF-based methods and 3DGS. Qualitatively, the novel-views from 3DGS-MCMC are sharp compared to the 3DGS method (Fig. 2). This is due to the MCMC formulation proposed in this work, which allows exploration.

- **High-Performance compared to 3DGS with a limited budget(L278-297):** The authors show an interesting experiment where they limit the budget for number of Gaussians during optimization. As expected, 3DGS has a significant drop in performance, whereas the performance drop in the proposed method is limited. Notably, there is a difference of 4 dB when the maximum number of Gaussians is set to 100k.

- Unlike 3DGS, the proposed method is not sensitive to the initialization. This robustness to initialization is illustrated in Tab. 2. When a camera extent of $1\times$ is used, 3DGS achieves a PSNR of 22.72 dB, whereas 3DGS-MCMC achieves a PSNR of 29.64. This result show that the proposed method is robust to the initialization. Also, this substantiates the exploration claim proposed in the paper.

- **Exhaustive ablation for all the key design choices:** In Tab. 3, the authors present ablation on the regularizers for Gaussians. Interestingly, in this framework, regularization of the Gaussian parameters improves the performance, whereas it is harmful in the 3DGS framework. Further, noise in the update step update allows for more exploration. This claim is further substantiated in Fig. 4. Finally, when noise is used for all the parameters, overall performance is slightly dropped.

**Weaknesses:**

- The proposed method exhibits robustness to the initialization step. However, is there a reduction in training time between the version using SFM points and the version using random initialization?

- **Missing evaluation on Scannet++[A1] dataset:** It is a large-scale dataset for indoor scenes. It will be interesting to see how this method performs on this challenging dataset. It is difficult to perform this experiment in a short duration of time. I leave it to authors to decide if they want to include in their manuscript.

- **Training Time:** The proposed method generates very high-quality novel-views. However, this comes with an added training cost. 3DGS-MCMC takes 90 minutes for 1M Gaussians, whereas 3DGS can be optimized within 30 minutes. The authors mention in L467-468 that a CUDA implementation can accelerate the proposed method.

[A1] Yeshwanth, C., Liu, Y.C., Nießner, M. and Dai, A., 2023. Scannet++: A high-fidelity dataset of 3d indoor scenes. In Proceedings of the IEEE/CVF International Conference on Computer Vision (pp. 12-22).

**Questions:**

- L40-41: "lead to .... waste compute."Can "poor-quality renderings" be substantiated by some examples? Also, can the authors elaborate why it is a wasted compute? InstantSplat[7] reconstructs an unbounded scene with sparse views in under 40 seconds on a commercial GPU.

- Does this method accurately represent the surface better than 3DGS? Can we extract a high-quality mesh from 3DGS-MCMC? Will it be better than recent works such as 2D Gaussian Splatting[A2]?

[A2] Huang, B., Yu, Z., Chen, A., Geiger, A. and Gao, S., 2024. 2d Gaussian splatting for geometrically accurate radiance fields. arXiv preprint arXiv:2403.17888.

**Limitations:**

The authors have discussed the limitations of their method in Appendix D in the supplementary material.
As such, this work has no societal impact. However, some downstream applications can have a societal impact. The authors have discussed this in Appendix E.

---

> ### Author Rebuttal · Authors · 2024-08-06
>
> We thank the reviewer for sharing the enthusiasm that we have for our method.
> Here is our response to your comments:
>
> ## Training time between SfM points version vs Random points version
>
> SfM has faster convergence, but to obtain the best PSNR performances both were run for about the same amount of wall-clock time. However, note our primary objective was not to improve convergence, but to reduce reliance on good initialization and also remove the heuristics. We believe the latter two to be critical in situations where initialization could be difficult, such as 3D generative modeling or dynamic scene reconstruction.
>
> ## ScanNet++
>
> Indeed this would be very interesting. We were not able to run these experiments due to the resource crunch during the rebuttal period, but we hope to be able to add these results for the camera ready.
>
> ## Training time
>
> Please see the global response for the training time. In short, 90ms was *milliseconds, not minutes* and considering only the compute time for each Gaussian the added time is negligible. Ours does run slower if trained with a high opacity regularizer due to more transparent Gaussians, but it ultimately gives higher quality results, and it also can be tuned to **train faster while still outperforming 3DGS**.
>
> ## L40--L41 Suboptimal placements leading to wasted compute and poor quality renderings
>
> Our existing experiments show that existing relocation heuristics place Gaussians at worse locations than ours, leading to lower rendering quality (this is especially true when random initialization is used). In regards to *wasted compute*, we wanted to highlight the fact that Gaussian splitting heuristics becomes unnecessary, but we realize that this can be misinterpreted and will remove this comment.
>
> ## Surfaces
>
> This is a great suggestion. While we were not able to finish this during the short rebuttal period as it involves setting up a new evaluation pipeline, we will investigate this in the future.

---

> > ### Comment · Reviewer_jFyG · 2024-08-07
> >
> > I have reviewed the rebuttal and my additional comments are as follows:
> > - First, I initially misunderstood "ms" as minutes. My mistake. According to the provided table, the proposed method takes 12 minutes longer than 3DGS. However, if fewer Gaussians are used, for example, 300k, the method converges in 21 minutes with a similar PSNR to 3DGS. This demonstrates the high efficiency of the proposed method.
> >
> > - Secondly, I hope these results, along with any additional experiments, will be included in the supplementary materials of the final camera-ready version.

---

> > > ### Author Response · Authors · 2024-08-07
> > >
> > > Thank you for the comments! and thanks again for acknowledging the efficiency of our method.
> > >
> > > Regarding your second point, we will for sure include them in the camera ready or the supplementary (if we run out of space)
> > >
> > > Thanks,
> > > Authors

---

### Official Review · Reviewer_Cpa7 · 2024-07-16

**Soundness:** 3
**Presentation:** 3
**Contribution:** 2
**Rating:** 5
**Confidence:** 4

**Summary:**

The paper presents a simple and effective method to enhance the training of 3D Gaussian Splatting (3DGS). It offers two main contributions. First, it demonstrates that adding carefully designed noise to the Gaussian centers after each gradient step can boost the performance of 3DGS. This encourages more exploration, which is especially helpful when Gaussian centers are randomly initialized. Second, the method performs densification by replacing low opacity Gaussians with clones of Gaussians sampled through multinomial sampling of the "live" ones based on their opacity values. The parameters of the Gaussians are adjusted to minimize the impact on the rendering outcome. The method is tested on both synthetic and real datasets and shows better performance compared to the 3DGS baseline, regardless of whether the Gaussian centers are randomly or "SFM" initialized.

**Strengths:**

- The paper tackles an important problem (optimization, cloning and splitting strategies for placing Gaussians) and introduces relevant concepts and ideas to analyze it.
- The paper proposes a simple and easy-to-implement way to improve the optimization of Gaussian positions by encouraging more exploration.
- The proposed relocation strategy works effectively  and pairs nicely with the L1-regularizer on the Gaussians.
- **Evaluation**: The evaluation is performed on various types of scenes (Nerf Synthetic, MipNeRF 360, Tank & Temples, Deep Blending, OMMO) and performance is reported with and without "SFM" initialization. Reporting the average over 3 runs and the corresponding standard deviation  is a good practice for the reliability of the results.
- **Performance**: The provided results show that the proposed method improves over the 3DGS baseline. More importantly, the method obtains competitive results without initializing the Gaussians with SFM points.

**Weaknesses:**

- **Relevance of MCMC framework**: Adding noise to the parameters [1] or to the gradients [2][3] is a common practice in optimization  to escape from saddle points and find local minima. Since the training relies on momentum-based optimizers, I believe the proposed method is more closely related to Perturbed Gradient Descent methods [4][5] or noise injection methods  [1] [2] [3] than to MCMC methods. For a proper momentum based update, noise would typically be added to the momentum estimate instead of the parameters. This doesn't make the method any less relevant or novel. **However, it does make me question the relevance of the MCMC framework that the paper is built around. In which part of the analysis or the method is this  framework necessary ?**
- **The design of the noise term**:  One important contribution of the paper is the design of the noise term. It would be great to have more explanations and ablations about these design choices. Why is this particular choice important for the method? Additionally, how does it compare to simpler noise schedules such as the one in [2]?
- Concerning the update in equation 9 of the paper, please clarify what assumptions and approximations are needed for this derivation and in which case they are no longer valid.


[1] Mark Steijkvers.A recurrent network that performs a context-sensitive prediction task. 1996.

[2] Neelakantan et al.,Adding gradient noise improves learning for very deep networks. 2015

[3]  Deng et al., How shrinking gradient noise helps the performance of neural networks. 2021.

[4] Jin et al., How to Escape Saddle Points Efficiently. 2017.

[5] Jin et al., On Nonconvex Optimization for Machine Learning: Gradients,
Stochasticity, and Saddle Points.  2021.

**Questions:**

- Relevance of the MCMC framework (See Weaknesses).
- How is the term designed and how does it affect the performance?
- What assumptions and approximations are needed for the derivation of equation 9 ?

**Limitations:**

The authors adequately addressed the limitations in the Appendix.

---

> ### Author Rebuttal · Authors · 2024-08-06
>
> We thank the reviewer for their constructive comments and suggestions.
>
> ## Relevance of the hybrid MCMC framework
>
> Our work is indeed related to gradient descent methods with noise and perturbations as suggested. In our case, our motivation for opting to interpret our framework as a hybrid MCMC method is due to the necessity of “jump” moves, and because we simply are much more familiar with the MCMC literature than what you pointed us to. As you suggest, we will extend our related works with the suggested literature in a revision. However, note that because much of the modeling space (the scene) is empty and Gaussians should only be located near surfaces, it is critical that Gaussians are moved toward these surfaces quickly. Under the MCMC formulation, they can be naturally integrated as “jump” moves or “resampling” moves [A, B], which we leverage.
>
> Regarding the noise in the momentum component, this is clearly an interesting direction to explore. We are excited that our reformulation seems to be opening doors to these various design choices which were originally not possible due to the heuristics that were involved in the optimization, such as abrupt reset of Gaussian opacities or manual splitting of large Gaussians. These heuristics clearly break any hope to formally study the convergence characteristics of the method, and in our work we remove all of them, which will hopefully lead to more work on this topic.
>
> [A] Lindsey et. al,  “Ensemble Markov chain Monte Carlo with teleporting walkers”, SIAM/ASA Journal on Uncertainty Quantification. 2022
>
> [B] Green, “Reversible jump Markov chain Monte Carlo computation and Bayesian model determination”, Biometrika, 1995
>
> ## Design of the noise term
>
> In Table 3 we already had a brief comparison of an alternative noise strategy where we add noise also to other Gaussian parameters. We have further tested without using the covariance term and the opacity term for the *Tank and Temples* dataset (we chose this dataset due to the short amount of time available for the rebuttal). We report the results in the table below:
>
> |              | No Covariance | No opacity* | With both |
> | ------------ | ------------- | ---------- | --------- |
> | PSNR - Train | 21.29         | 21.68      | 22.40     |
> | PSNR - Truck | 25.04         | 23.27      | 26.02     |
> | PSNR - Avg.  | 23.16         | 22.47      | 24.21     |
>
> Having both covariance and opacity is important to achieve the best performance:
> - Without the covariance term, the noise is difficult to “undo” with gradients for narrow Gaussians, which then cause the Gaussians to randomly walk about and no longer represent the desired distribution.
> - Without the opacity term, opaque Gaussians that are significantly contributing to the reconstruction loss would not be able to converge/stabilize, leading to a loss of high-frequency details.
>
> Further, note that in the table above we had to use $\lambda_{noise}=5\times10^2$ for “No opacity*”, as if we use the same hyper-parameters as in the paper, the model does not train at all (PSNR<7).
>
> ## Noise term scheduler
>
> As suggested, we also tried the scheduling from [2], and we additionally tested a simple linear scheduler. Again, we ran our experiment on the *Tank and Temples* dataset. As shown in the table below, our exponential decay scheduler (which uses the same scheduler as in the original 3DGS paper) works the best:
>
> |              | Noise Scheduler in our paper | Best Noise scheduler in [2] | Linear scheduler |
> | ------------ | ---------------------------- | --------------------------- | ---------------- |
> | PSNR - train | 22.40                        | 21.86                       | 15.37            |
> | PSNR - Truck | 26.02                        | 23.06                       | 19.90            |
> | PSNR - Avg.  | 24.21                        | 22.46                       | 17.64            |
>
> ## Derivation of Eq. 9
>
> The detailed derivation of Eq. 9 is provided in Appendix. A. The main approximation is the approximation of the integral that integrates the squared difference between the two distributions. We instead minimize the difference between the individual integrals, projected into each view, which is inspired by sliced Wasserstein methods [17]. As the color values that we integrate over are always non-negative, there are no underlying assumptions that could break.

---

> > ### Comment · Reviewer_Cpa7 · 2024-08-14
> >
> > Thank you for the responses. The authors have addressed my concerns in the rebuttal. I think the noise ablation should be included in the  paper.
> >
> > Concerning the relevance of the hybrid MCMC framework, I understand how the relocation of Gaussians can be seen as  “jump” moves [B]. However, I don't see how this makes sure that "Gaussians are moved toward these surfaces quickly" since the  “jump” moves happen between "states" which depends on how the reloacation is performed.
> >
> > Overall, I think this is a good contribution and the authors did a good job addressing my concerns.  I have decided to maintain my rating in support of acceptance.

---

### Author Rebuttal · Authors · 2024-08-06

We are glad to see that all reviewers are positive towards our paper.
Reviewers commend the effectiveness of our method (**Cpa7**, **jFyG**, **cJQR**), especially when random initialization is used, the thoroughness of our ablations (**jFyG**, **cJQR**).
They also acknowledge the ease of use of our method as a drop-in to existing Gaussian Splatting methods (**jFyG**, **cJQR**).
Please see the individual rebuttals for reviewer-specific responses.

## Training time (**jFyG** and **cJQR**)

First of all, we would first like to clarify that what we provided in the Appendix for training time is 90ms (milliseconds) and *not 90 minutes* (**jFyG**).

To provide an exact comparison, we took our Gaussians trained at 1M Gaussian, and re-measured the single optimization iteration time for our method and the original 3DGS. In this case, ours takes 80 milliseconds while 3DGS takes 76 milliseconds. That is, the added time for sampling and noise addition is not substantial, even with our implementation.

Still, to achieve the PSNR reported in the paper, our method does take longer. This is because the configuration of Gaussians (i.e. where they are, their sizes and opacity) matters greatly to the runtime as this affects the speed of rasterization (among them the opacity regularizer affects the speed the greatest). For the “room” scene in the MipNeRF 360 dataset, we find the following timings, all with SfM initializations and max 1.5M Gaussians per the original 3DGS implementation:

|              | opacity regularizer ($\lambda_o$) | PSNR | Total Training Time |
| ------------ | ------------- | ---------- | ---------- |
| 3DGS | --  |31.7 | 25 minutes |
| Ours (paper) | 0.01 |  32.5 | 42 minutes |
| Ours  | 0.001 |  32.4 | 30 minutes |
| Ours (300k Gaussians)  | 0.01 | 31.8 | 21 minutes|

Note how our method still **outperforms 3DGS regardless of our choice**. We found $\lambda_o=0.01$ to work well for various scenarios, including when using random initialization (with lower opacity regularizer random initialization performs slightly worse 31.7 vs 32.3 with $\lambda_o=0.001$), thus we report performance with 0.01. Finally, when using fewer Gaussians, our method **still outperforms 3DGS and trains faster**.
Due to the scarcity of compute caused by the rebuttal period, we were unable to provide the full per-scene compute times, but we will include them all in our camera ready.

---

### Decision · Program_Chairs · 2024-09-25

**Decision:**

Accept (spotlight)

**Comment:**

The submission received consistently high scores. The reviewers and the authors engaged in positive discussion during the rebuttal phase, and a few of the initial queries were satisfactorily resolved (the biggest one was a misunderstanding of ms wrt time). The proposed solution was found to be simple and effective, removing the heuristic step in the original GS initialization/spawning step. The work is appreciated both due to its theoretical grounding and its practical relevance.

Please include the additional clarifications while preparing the final version of the paper. Congratulations on the nice piece of work.